# Experimental study on polymer friction composite with natural friction modifiers for brake pads

Mitali Gore[1], Ajit Bhosale[2]*, Mithul Naidu[3], Robert Čep[4], Sachin Salunkhe[5,6], Emad Abouel Nasr[7]*

1 Zeal College of Engineering and Research, Pune, Maharashtra, India, 2 Department of Mechanical Engineering, MKSSS's Cummins College of Engineering for Women, Pune, Maharashtra, India, 3 Department of Mechanical Engineering, Tolani Maritime Institute Induri, Pune, Maharashtra, India, 4 Department of Machining, Assembly and Engineering Metrology, Faculty of Mechanical Engineering, VSB-Technical University of Ostrava, Ostrava, Czech Republic, 5 Department of Biosciences, Saveetha School of Engineering, Saveetha Institute of Medical and Technical Sciences, Chennai, Tamil Nadu, India, 6 Department of Mechanical Engineering, Gazi University Faculty of Engineering, Maltepe, Ankara, Turkey, 7 Department of Industrial Engineering, College of Engineering, King Saud University, Riyadh, Saudi Arabia

* ajit.bhosale@cumminscollege.in, eabdelghany@ksu.edu.sa

## Abstarct

The increasing concern over environmental pollution from brake dust and the adverse impacts of conventional brake pad materials, such as metallic, semi-metallic, and ceramic composites, has prompted the exploration of more sustainable alternatives. Traditional brake pads release harmful non-exhaust emissions that contribute to air pollution and wear down quickly, posing both environmental and operational challenges. This study investigates the development and performance evaluation of polymer friction composites enhanced with natural friction modifiers sourced from agricultural waste materials like walnut shell, coconut shell, and groundnut shell powders. These materials were selected for their biodegradability, lightweight properties, and cost-effectiveness. Three types of polymer reinforced composites—WPRC (Walnut Powder Reinforced Composite), CNPRC (Coconut Shell Powder Reinforced Composite), and GNPRC (Groundnut Shell Powder Reinforced Composite)—were manufactured according to ASTM G99 standards. A pin-on-disc tribometer or an L27 Taguchi experimental setup were used to measure friction and wear. The tests showed that WPRC had the best general performance of all the composites that were tried. It had stable coefficients of friction and a low specific wear rate. The results indicate that natural friction modifiers could improve the environmental and operational performance of polymer composites, by replacing existing brake pad materials with a more environmentally friendly alternative.

**Data availability statement:** All relevant data are within the paper.

**Funding:** The authors present their appreciation to King Saud University for funding this research through the Ongoing Research Funding program (ORF-2025-164), King Saud University, Riyadh, Saudi Arabia. This article was co-funded by the European Union under the REFRESH – Research Excellence For Region Sustainability and High-tech Industries project number CZ.10.03.01/00/22_003/0000048 via the Operational Programme Just Transition and has been done in connection with project Students Grant Competition SP2024/087 Specific Research of Sustainable Manufacturing Technologies "financed by the Ministry of Education, Youth and Sports and Faculty of Mechanical Engineering VŠB-TUO. The article has been done in connection with the project Students Grant Competition SP2024/087", Specific Research of Sustainable Manufacturing Technologies "financed by the Ministry of Education, Youth and Sports and Faculty of Mechanical Engineering VŠB-TUO.

**Competing interests:** The authors have declared that no competing interests exist.

## 1. Introduction

The global prohibition of asbestos fibers in the late 1980s represented a key moment in the development of brake pad materials, as it compelled manufacturers to find safer substitutes for the toxic material [1,2]. Asbestos once formed the basis for brake systems due to its excellent frictional characteristics, making braking efficient even in extreme conditions [3]. As effective as it was, it was subsequently determined to have significant carcinogenic properties which put both manufacturers and consumers at a serious risk to their health [4]. This recognition led to a worldwide transition to non-asbestos organic (NAO) friction materials which were deemed safer and more eco-friendly [5]. NAO materials which included synthetic fibers like glass, carbon, and aramid, became popular very quickly [6]. These fibers, while offering excellent thermal stability and durability, were chosen for reinforcing friction composites in brake pads [7]. Nevertheless, the increased focus on environmental sustainability in recent decades has revealed the limitations of synthetic fiber-reinforced polymers (FRPs), especially in the automotive sector [8]. Although synthetics perform well, they have inferior recyclability and disposal (due to end of life) issues in terms of environmental hazards. Synthetics are not biodigradable, causing negative and for long term environmental degradation. Those enduring behaviours associated with degradation and disposal have also led to higher regulations [9]. A pleasant range of frameworks have been adopted for these materials, such as the EU Landfill Directive (Directive 1999/31/EC) and the End-of-Life Vehicle Directive (Directive 2000/53/EC), that aim to reduce the ecological footprint of such materials [10]. These regulations implement strict controls on disposal for automotive components including brake pads, which has expedited the search for alternative, more sustainable materials [11]. In addition, an additional obstacle is the high costs of synthetic fibers, which is particularly alarming for an industry that is fundamentally based on low-cost items. This incredibly limiting factor only serves to further necessitate the exploration of natural, biodegradable options that will act as substitutes for synthetic reinforcements in brake lining formulations [12].

A lot of attention has been paid to natural fibres, especially bast fibres like flax, jute, kenaf, and ramie, as possible alternatives to synthetic fibres in a number of vehicle uses, such as brake pads [13]. These fibres can be used again and again, break down naturally, and are much better for the earth than manufactured materials [14]. Furthermore, natural fibers also offer better mechanical properties, demonstrated by their high specific strength and low density, and can be used in brake pad formulations requiring better performance and durability [15]. Utilizing natural fibers in friction materials is more than a stride towards sustainability, it is a pragmatic means to absolve the automotive industry from their source of non-renewable materials. This evaluation studied 3 different natural fiber reinforced polymer brake pad formulations which, were formed and characterized structurally, as well as formulated with natural friction modifiers to improve the tribological performance of the materials [16]. These materials are called WPRC, CNPRC, and GNPRC, and they were made better using the Taguchi design of experiments (DOE) method, which carefully figures out the best combinations of parameters for each brake pad formulation. It gives a lot of

information on how the processing variables affect performance characteristics like coefficients of friction (COF) as well as particular wear rate (SWR) [17].

There is a popular and proved way to make hard processes better, and the technique developed by Taguchi can be used to make brake pad materials work better in tribology. This lets researchers look at how different things (like fibre type, friction modifiers, and even manufacturing settings) can affect how well brake pads work [18]. An analysis of variance (ANOVA) method was used again in this study to look at the functionality of the brake pad formulas from the edited process used in earlier chapters. ANOVA is the statistical method that measures the amount of instrumental variable effects on friction stability and wear rate. In this sense, the researchers will determine which of the parameters can change the breaking pad's performance characteristics into those of interest, and then they will apply automatic combinations of the best outcome parameters to stretch out the breaking pads overall performance functional area. Based on the previous hypotheses demonstrated in the documented performance in the prior chapter, the preliminary results of the study show that the performance of the WNPCR variation is better than the other variations in terms of friction stability and wear rates, while providing an alternative to conventional brake pad materials that meet eco-friendly criteria that can be classed as environmental sustainable alternatives [19]. Utilization of natural fibers, as well as the concept of optimization, lends itself to a great opportunity for establishing high-performing ecological brake pad solutions to better address both environmental and economic issues. The current research contributes to the information and knowledge of sustainable materials in the automotive industry, while providing a basis for future innovations in brake pads [20].

## 2. Materials and methods

### 2.1 Fabrication of composition

Natural fillers like walnut shell powder form, coconut shell powder, and peanut shell powder are used in eco-friendly polymer-based friction compounds in this study. The intended use of natural fillers have various properties: walnut shell powder provides excellent frictional stability and rigidity, coconut shell powder provides thermal insulation and has a low density, and groundnut shell powder provides abrasion resistance and is cost-effective. The binder (20% by weight) was phenol-formaldehyde resin, which will deliver structural integrity while providing resistance to thermal degradation. Barium sulfate ($BaSO_4$) was the predominant filler at 40% by weight, but it also provides improvements in thermal conductivity and mechanical strength. Graphite (5%) was added as a solid lubricant to improve tribological performance as it lowers the coefficient of friction during braking. Vermiculite (5%) was added as thermal stabilizer which will enhance both heat resistance and mechanical performance [21].

The weight ratio of natural fillers was determined according to tribological performance benchmarks, such as the level identified by Mithul Naidu et al. (2022) and 30% natural filler content in the study of them indicated such content was best for frictional and wear properties of similar composites. The composite formulation used combines thermal, mechanical, and tribological integrated characteristics that meet the requirements of brake pads, such as sustainability and cost-effectiveness. The material composition is shown in Table 1, and the proportion of the relevant components were carefully empirically charted to formulate a high-performance brake pad material. The composite produced were later subjected to testing for wear resistance, frictional behaviour and thermal testing against different operating conditions to assess suitability for automotive applications [22].

The polymer friction composites for brake pad applications were prepared using the standard and precise methods to obtain proper performance. The parts were weighed with a digital scale from Wensar® weighing scales Ltd in Chennai, India, which was accurate to within 0.01 g and could measure between 0 and 220 g. To get a smooth mix, the ingredients, crushed fibres, and phenolic epoxy powder were mixed with a motorised mixer at 250–500 rpm for 15 minutes. The blend was then compression molded using a Santec Inc. (Delhi, India) compression molding machine [23]. The molding procedure was conducted under controlled conditions at 155°C and 15 MPa for a period of 10 minutes. During the molding

**Table 1. Material compositions in weight percentages.**

| Materials | Weight Contribution (%) | | |
|---|---|---|---|
| | WFRC | CPRC | GPRC |
| walnut powder | 30 | 0 | 0 |
| coconut powder | 0 | 30 | 0 |
| groundnut powder | 0 | 0 | 30 |
| Phenol Formaldehyde | 20 | 20 | 20 |
| Graphite powder | 5 | 5 | 5 |
| Vermiculite | 5 | 5 | 5 |
| Barium sulphate | 40 | 40 | 40 |
| Total | 100 | 100 | 100 |

process, four venting cycles were introduced to help facilitate composite formation. Following the compression moulding process, post-curing was carried out for an hour at 100°C in an oven with hot air (Athena Technologies, Mumbai, India). The post-curing step was deemed necessary for removing residual moisture and vapours that may have been trapped during the polymerization process and for relieving compressive stresses associated with molding [24].

The finished composite plates were produced in sizes of 100 x 100 x 10 mm from three distinct materials, WPRC, CNPRC, and GNPRC, manufactured using the compression molding process, then cut into specimens for the wear-testing process in accordance with ASTM G99 procedures. The wear performance was evaluated using the Pin-On-Disc wear test, as demonstrated in Fig 1a–c (not provided here) [1]. The weight percentages are chosen empirically to balance the tribological properties (friction and wear resistance). These proportions were chosen empirically so that there was a sufficient balance of friction, wear resistance, and thermal stability to make sure the composites can be used as brake pads.

### 2.2 Friction and wear testing

#### 2.2.1 Design of experiment using orthogonal array.
This study investigates the frictional behavior of polymer-based friction composites for brake pads using Taguchi's experimental design approach. This approach is well known for its efficiency in discovering the best process parameters with the least number of experimental trials. The authors identified three composites, including a Polymer Resin Composite (WPRC), CNPRC, GNPRC. There were a lot of factors that went into the tests, including the Correlation of Friction (COF), the Specific Wear Rate (SWR), and the different working situations. The experiments were with the Taguchi L9 orthogonal array design procedure and enabled all three parameters (Material type, applied load and sliding speed) to be varied systematically [25]. All three parameters were established at three levels to maximize the knowledge of the tribological behaviour of the composites. This orthogonal array design procedure reduced the total number of experimental runs while increasing the efficacy while analysing the results Table 2.

The L9 orthogonal array made it easy to study not only the main effects of the known factors, but also how they affected the COF and SWR when they worked together. It was found that natural friction stabilizers had a big impact on the composition's wear resistance or friction stability, especially in the GNPRC composite materials. Additionally, the body of data obtained provided a performance trend consistent with the predicted nature of advanced friction composites when braking. Each test condition was done three times, with new specimens used for each load-speed-composition set up. This was done to make the statistical analysis more reliable. Replicating the test conditions in this way allowed for an assessment of variability, and increased the confidence in the results presented. In summary, this project demonstrates the merit of applying Taguchi's design process in the optimization of composite brake materials and provides meaningful founding blocks for the efficient design of high-performance, durable, and efficient friction materials for automotive braking systems [26].

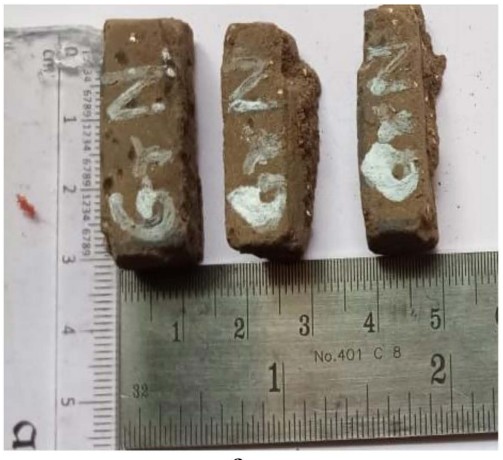
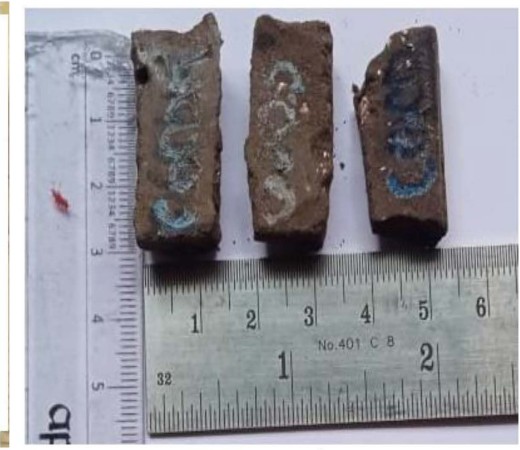

a

b

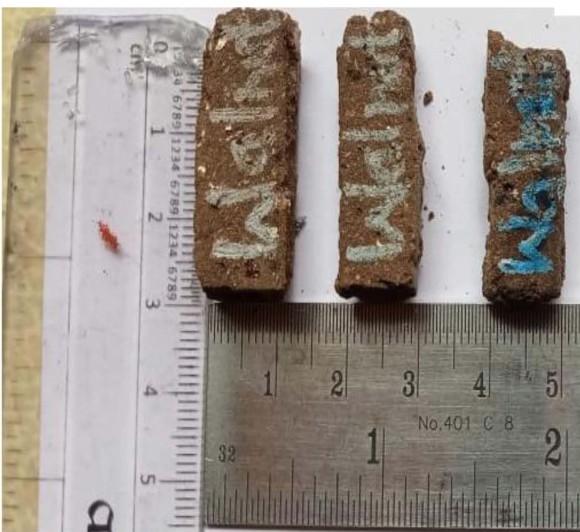

c

**Fig 1. (a) WPRC, (b) CNPRC, (c) GNPRC.**

**Table 2. Factors and levels of Phase 1 composites for Taguchi D.O.E.**

| Factors | Units | Level 1 | Level 2 | Level 3 |
|---|---|---|---|---|
| Composition | – | WPRC | CNPRC | GNPRC |
| Load | N | 30 | 50 | 70 |
| Sliding Velocity | m/s | 2.6 | 3.9 | 5.2 |

**2.2.2 Experimental procedure.** A study using experiments was done to see how well three polymer-based frictional composites worked as brake pads. The tests were done on a Pin-On-Disc machine (DUCOMTM TR20LE, Bohemia, NY, USA) following the ASTM G 99 guidelines to make sure that the settings were standard and the results were uniform. To get a better idea of how things wear down in real life, a 100-mm-wide track that has a 5000-meter slide a distance was used. A Taguchi L9 orthogonal array with multiple variables at three levels was used to improve the trial design. This

design allowed for an efficient and systematic evaluation of multiple combinations of parameters without the necessity of a large number of experiments. The primary goal was to determine the wear characteristics and tribological performance of each composite under different test conditions. Thus, the researchers were able to assess the have an influence on performance indicators, such as wear and friction [27].

During the testing stage for each composite sample, key tribological properties of interest (Friction coefficient (COF), specific wear rate (SWR), etc.) were examined. Each sample was tested under each testing condition three times to provide uniformity. When comparing properties, the average value for the property was used in lieu of the sample values to lessen the effect of random variability among trials. Each of the numbers in Table 3 was used as its own individual factor to judge the performance of the combined material. ANOVA, a statistical tool, was used to check the data for significant and make sure that the changes seen between factor levels were based on true statistical traits. The raw data was also shown. The full test logs, including COF and SWR values, load and wear data for each tester sample were retained as they occurred. This adds to transparency and reproducibility and can be confirmed by a third party [28]. Over the course of this project, a substantial amount of valuable information was documented that would be of benefit to the field of high-performance friction materials (wear resistant, friction materials).

To support the statistical robustness of the results, I also included extra statistical analysis. In addition to the p-values reported in both Tables 4 and 5, I computed standard deviations and 95% confidence intervals (CIs) for SWR and COF. Standard deviation (SD) values were calculated from triplicate experimental measurements for each test condition in order to account for variability within replicates [29]. For example, under a load of 30 N, WPRC showed a SWR of $0.7428 \pm 0.012$ mm³/Nm, meaning there was low variability and robust reliability of measurement. The same could be said for the COF of GNPRC at 70 N with a $0.903 \pm 0.008$ value demonstrating strong frictional consistency.

The 95% confidence intervals for each performance metric were computed using the formula:

$$CI = Mean \pm (t \times (SD/\sqrt{n})),$$

**Table 3. Specific wear rate (SWR) and coefficient of friction (COF) using L9 orthogonal array.**

| Composition | Sliding Velocity (m/s) | Load (N) | SWR (mm³/Nm × 10⁻⁶) ± SD | COF ± SD |
|---|---|---|---|---|
| WPRC | 2.6 | 30 | $0.7428 \pm 0.015$ | $0.486 \pm 0.008$ |
| WPRC | 3.9 | 50 | $1.4026 \pm 0.021$ | $0.484 \pm 0.006$ |
| WPRC | 5.2 | 70 | $0.9710 \pm 0.018$ | $0.481 \pm 0.005$ |
| CNPRC | 2.6 | 30 | $1.3210 \pm 0.026$ | $0.431 \pm 0.010$ |
| CNPRC | 3.9 | 50 | $3.1100 \pm 0.032$ | $0.342 \pm 0.009$ |
| CNPRC | 5.2 | 70 | $2.4870 \pm 0.030$ | $0.450 \pm 0.007$ |
| GNPRC | 2.6 | 30 | $0.3730 \pm 0.012$ | $0.574 \pm 0.010$ |
| GNPRC | 3.9 | 50 | $0.5380 \pm 0.014$ | $0.581 \pm 0.011$ |
| GNPRC | 5.2 | 70 | $0.2951 \pm 0.011$ | $0.903 \pm 0.009$ |

**Table 4. Comparison of developed composites with commercial pads.**

| Property | WPRC (Best Natural) | Semi-metallic Pad | Ceramic Pad |
|---|---|---|---|
| Coefficient of Friction (COF) | Low | Moderate | Moderate |
| Specific Wear Rate (SWR) | High | Moderate | Low |
| Thermal Resistance | Moderate | High | Very High |
| Environmental Impact | Low | High | Medium |
| Cost | Low | High | High |
| Fade Resistance | Low | High | Very High |

**Table 5. ANOVA for S/N ratio of specific wear rate.**

| Source | DF | Seq SS | Adj SS | Adj MS | F | P | Contribution (%) |
|---|---|---|---|---|---|---|---|
| Composition | 2 | 359.877 | 359.877 | 179.939 | 44.66 | 0.022 | 83.05 |
| Load | 2 | 46.049 | 46.049 | 23.024 | 5.72 | 0.149 | 10.62 |
| Velocity | 2 | 19.315 | 19.315 | 9.658 | 2.40 | 0.294 | 4.45 |
| Residual Error | 2 | 8.057 | 8.057 | 4.029 | | | |
| Total | 8 | 433.298 | | | | | |

S = Sum of Squares in a certain order; Adj. SS = Sum of Squares that have been adjusted; Adj. MS = Adjusted Mean Square; What does DF stand for? It's written as F = Ratios of variance explained over unclear variance, P = The chance of getting the F value.

where t is the critical t-value for n-1 degrees of freedom at α = 0.05. This interval data indicated that variation in results across composite types was statistically significant and reliable across trials. The p-values reported in the ANOVA confirmed that composition was the main factor for both SWR and COF and accounted for more than 80% of the variation observed [30]. This confirms the seen trends were reliable and also can be compared between different formulations.

The data the authors generated and analyzed throughout this study consist of raw experimental logs, which include coefficients of friction, specific wear rate measurements, and repeated values during testing for each composite formulation (WPRC, CNPRC, and GNPRC) [31]. All of this data is important for validating the conclusion of the study and demonstrating reproducibility.

## 3. Results and discussion

The experiment was carried out in a systematic experimental way using Taguchi Design of Experiments (DOE) to investigate a polymer friction composite for brushless electric simulated brake pads. The experiment took into consideration three major engineering parameters in composite brake pad engineering: composition, load, and sliding velocity. Each of the three variables had three levels which allowed us to provide an L9 orthogonal array to efficiently establish and fix all factors. The composite's tribological behaviour was studied using a Pin-on-disc tribo-machine and the ASTM G99 standard wear test method. This gave a standard testing environment and a reliable way to measure wear features following a standard process. To find the Specific Use Rate (SWR) and Coefficient of Friction (COF), we used normal test assessor methods. This let us see how well the material behaved in terms of resistance to wear and friction. The Taguchi method says that the S/N ratios of the materials were used to figure out how to best guess what would happen so that there was the least amount of wear and the most friction. This was done by choosing the best mix of the three factors.

Using the S/N ratios, it was possible to identify the conditions for the polymer composite while achieving a compromise between frictional performance and wear. This excuse for a structured optimization study is significant so that you can create an advancement to brake pad material, because you want to achieve a composite that performs well, but is also tolerant to a range of conditions of application, applied loading and sliding velocity. This study generates information that is oriented specifically for producing improved advanced brake pad materials that function and last, resulting in safer braking systems and potential improvement in performance.

$$SWR = \frac{\Delta m}{\rho LD} \tag{1}$$

$$COF = \frac{F}{L} \tag{2}$$

*Where, $\Delta m$ = mass loss, $\rho$ = density, L = applied load, D = sliding distance = 5000m (constant), F = frictional force.*

## 3.2 Variation of friction force (F), SWR and COF with respect to normal load

The changing friction force over time has been seen for all three materials at 30 N, 50 N, and 70 N, as shown in Fig 2a–c.

When looking at polymer friction composite materials for brake pads, sharp wear and tear is a big problem that mostly shows up as a "ploughing effect." This phenomenon arises when larger particulates or small protrusions penetrate the brake pad material, causing the surface to be scraped off and the formation of abrasive detritus and wear scarring. When this load is put on the touch surface, these hard particles press into it and make depressions. These depressions make the friction pair's surface even less smooth, which causes tiny bulges to form on the contact points. The brake pad experiences adhesive degradation as a result of the bond points that are generated by the expanded contact area between these bulges [32]. When brake pads are used at high speeds and with a lot of weight on them, the contact peaks change shape a lot and the surface temperature rises. In turn, this causes stress at the bond sites, which causes cracks and material to be spread out, which creates rough grit. Some of these particles move from the pads surface to the disc surface, which makes the disc wear out faster. The experiment results shown in Fig 2a–c show a constant process of wear and material transfer. This process has two different types of friction: a running into period and a steady-state time.

The glue wear process makes the bond between the test elements and the metal blade stronger during the running-in time. Subsequently, the system undergoes a steady-state phase, during which friction forces remain constant over time. The friction forces during this phase are consistent, as evidenced by the fact that this steady-state period remains stable under all three normal load conditions WPRC, CNPRC, and GNPRC. Friction forces are directly correlated with the normal stresses imparted to the friction surface, as demonstrated by the results. The friction forces increase in proportion to the increase in the normal load [33]. Figs 3 and 4 show the testing results for the Sliding Wear Rate (SWR) and The coefficient of Friction (COF) at three different average load levels: 30 N, 50 N, and 70 N. There is a clear trend in these images that shows friction forces rise as the average load rises. Please keep in mind that the slide speed wasn't considered in these experiments because it wasn't thought to have much of an effect on the friction reaction, as shown by the numbers in Tables 4 and 5.

Fig 3 shows the unique wear rate (SWR) of three different polymer friction composites: WPRC, CNPRC, and GNPRC. The information shows that CNPRC's SWR is much lower than those of GNPRC and WPRC. Also, as the average load goes from 30 N to 70 N, the SWR goes down for all three combinations. Equation (1) shows that this is because SWR is opposite to the average load. It is mostly adhesive wear at low normal pressures, where fibres and matrix materials stick together and come into close touch with each other. The process alters to attrition and a rough wear form when the usual load is raised [34]. Because of the heat that build up when the friction blend touches the metal counter face, this change takes place. This means that as loads rise, the wear properties of polymer friction composites will change. As normal stresses rise, they will go from sticky wear to abrasive wear. This shows that the load pattern has a big effect on the wear properties of brake pads.

The coefficient of Friction (the COF) numbers for the three different compositions—CNPRC, WPRC, and GNPRC—are shown in Fig 4. These values show clear trends as the standard load goes from 30 N to 70 N. It is seen that the COF goes down as the average load goes up for CNPRC, but it goes up for both WPRC and GNPRC. Equation (2) shows the mathematical relationship that this behaviour follows. The drop in COF for CNPRC could be because of changes on the worn surface, possibly because a transfer layer formed on the friction surface. Another thing that might have caused the drop in COF is an increase in the asperity contact temperature when the load is higher. Among these compositions, GNPRC stands out by maintaining lower and more stable COF values. This can be explained by its superior fiber-matrix interfacial bonding, which is more robust compared to the other two compositions, WPRC and CNPRC, resulting in better wear resistance and reduced frictional variations.

Table 4 compares the characteristics of three forms of brake pads, namely WPRC (Best Natural), semi-metallic, and ceramic pads. The Coefficient of Friction (COF) for WPRC was low but moderate for the semi-metallic and ceramic pads.

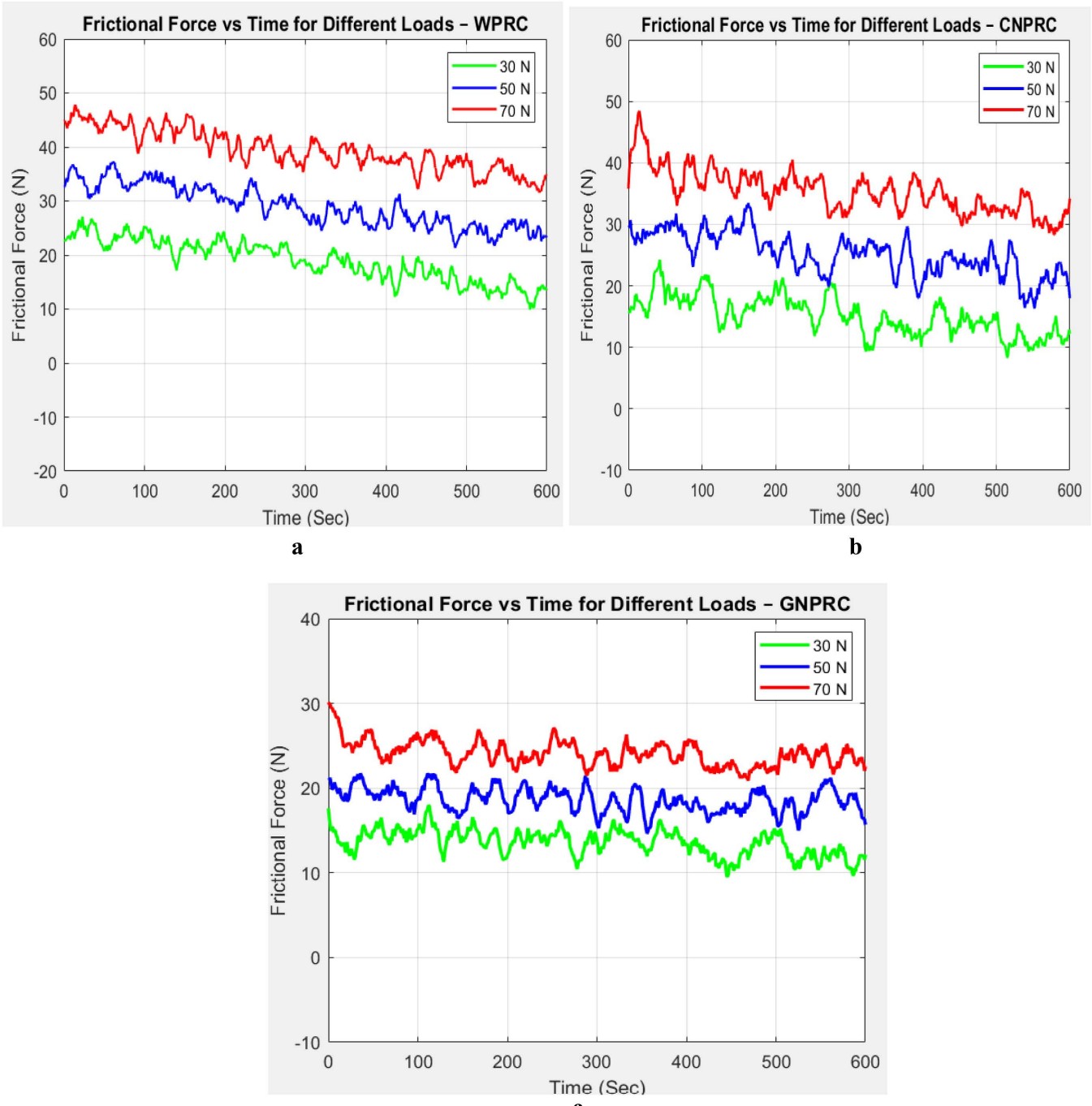

**Fig 2. Variation of friction force with respect to time for (a) WPRC, (b) CNPRC, (c) GNPRC.**

This means all pads generated friction levels appropriate for stopping but the semi-metallic and ceramic pads would have potentially better control. The Specific Wear Rate (SWR) for WPRC was high, meaning it was getting more wear than the other two pads, with the ceramic pad getting the least. The Thermal Resistance was moderate in WPRC, high in

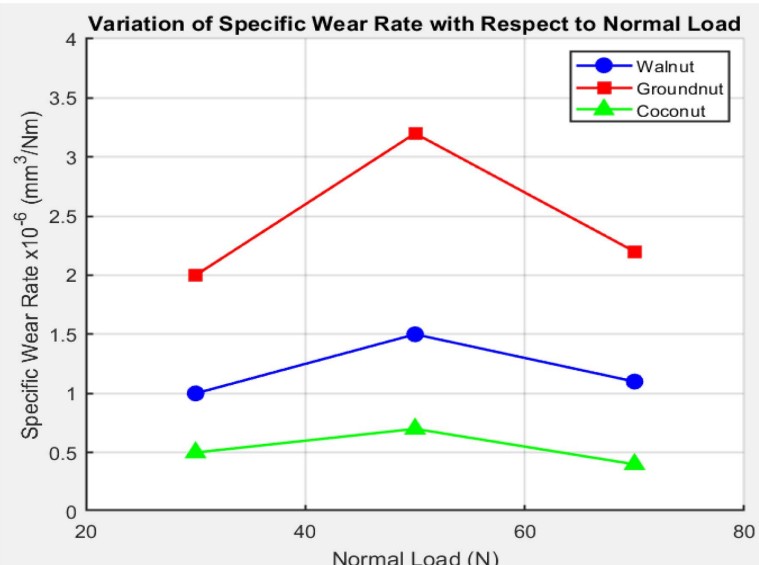

**Fig 3. Variation of specific wear rate with respect to normal load.**

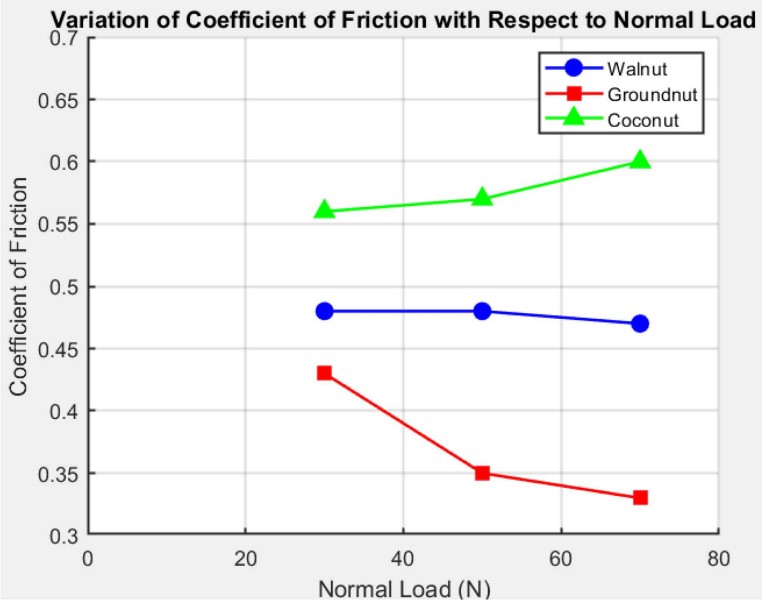

**Fig 4. Variation of coefficient of friction with respect to normal load.**

semi-metallic pads, and very high in ceramic pads. This indicates the ceramic pads are more suited for high-temperature applications and, therefore, less susceptible to thermal degradation.

When it comes to Environmental Impact, WPRC has the least environmental burden, with a low impact, while semi-metallic pads have a high environmental impact due to the materials used in their composition. Cost is another significant difference, with WPRC being the most economical option, while both semi-metallic and ceramic pads are more expensive.

Lastly, the Fade Resistance of WPRC is low, meaning it is less resilient under repeated stress, while ceramic pads exhibit very high fade resistance, making them ideal for heavy-duty applications. This comparison provides valuable insights into the trade-offs between performance, cost, and environmental impact of each material.

### 3.3 Analysis of variance

You can use Analysis of Variance (ANOVA), a powerful mathematical tool, to find out how independent factors like composition, load, and motion affect performance measures that are based on those variables, such as Specific Wear Rate (SWR) or Coefficient of Friction (COF). The method systematically examines the relative significance of each factor by computing F-values, which indicate how strongly a factor affects the outcome. A higher F-value corresponds to a more pronounced effect, whereas a lower value indicates minimal or statistically insignificant influence. The critical benchmark for significance is typically the 5% level ($p < 0.05$); factors falling below this threshold are regarded as insignificant. ANOVA also tells you how much each factor contributed by dividing its own sum of squares by the total sum of squares. This percentage addition shows how important each variable is in explaining the observed variation. It's important to note that this study's analysis included three trials for each scenario. This made it possible to get a good idea of the experimental error and made the results more reliable.

The ANOVA results for SWR or COF are shown in Tables 5 and 6, which show how each factor affects the total range. For SWR, the normal load emerges as the most influential factor, accounting for 83.05% of the total variation. This is followed by composition with 10.62%, and sliding velocity with a lesser 4.45%. Conversely, the analysis of COF reveals a different dominance pattern, with composition contributing the most at 81.85%, followed by normal load at 7.52% and velocity at 6.82%. These results demonstrate that while load predominantly governs material wear, the material composition has a more significant impact on friction behavior. Such insights are critical for optimizing material and operational parameters in tribological applications.

### 3.4 Optimization of factors

The Taguchi design method was used to evaluate the 8-levels of factors affecting Specific Wear Rate (SWR) and Coefficient of Friction (COF) and we obtained the optimal levels based on the results for the main effects plots for Signal-to-Noise (S/N) ratios [35]. In Figs 5 and 6, you can see the best amounts of factors for SWR and COF. In both SWR and COF tests, the plots show that Walnut The powder Composite Composite (WPRC) with a load of 70 the Newtons and a moving speed of 2.6 m/s is the best combo. If you want to minimise SWR while increasing COF, this mix is the best; thereby delivering the best tribological performance under the experimental conditions examined in this work. The outcomes of this study provide evidence on the importance of these factors for improving the overall performance of brake pad materials.

The previous statistical visualizations illustrate three levels, and provide the different composition, load, and velocity values as the highest and lowest values for the parameters diverge. The individual composition values are larger for Level

**Table 6. ANOVA for S/N ratio of coefficient of friction.**

| Source | DF | Seq SS | Adj SS | Adj MS | F | P | Contribution (%) |
|---|---|---|---|---|---|---|---|
| Composition | 2 | 15.2721 | 15.2721 | 7.6361 | 21.54 | 0.044 | 81.85 |
| Load | 2 | 1.2732 | 1.2732 | 0.6366 | 1.8 | 0.358 | 6.82 |
| Velocity | 2 | 1.4036 | 1.4036 | 0.7018 | 1.98 | 0.336 | 7.52 |
| Residual Error | 2 | 0.7089 | 0.7089 | 0.3544 | | | |
| Total | 8 | 18.6578 | | | | | |

While this BNG (Basic Number Generator) is limited in several respects, even the BNG can yield an acceptable baseline for travelers to use while traveling. The BNG technique is one possibility. Travelers have many choices they can avail of in their experiences of travel.

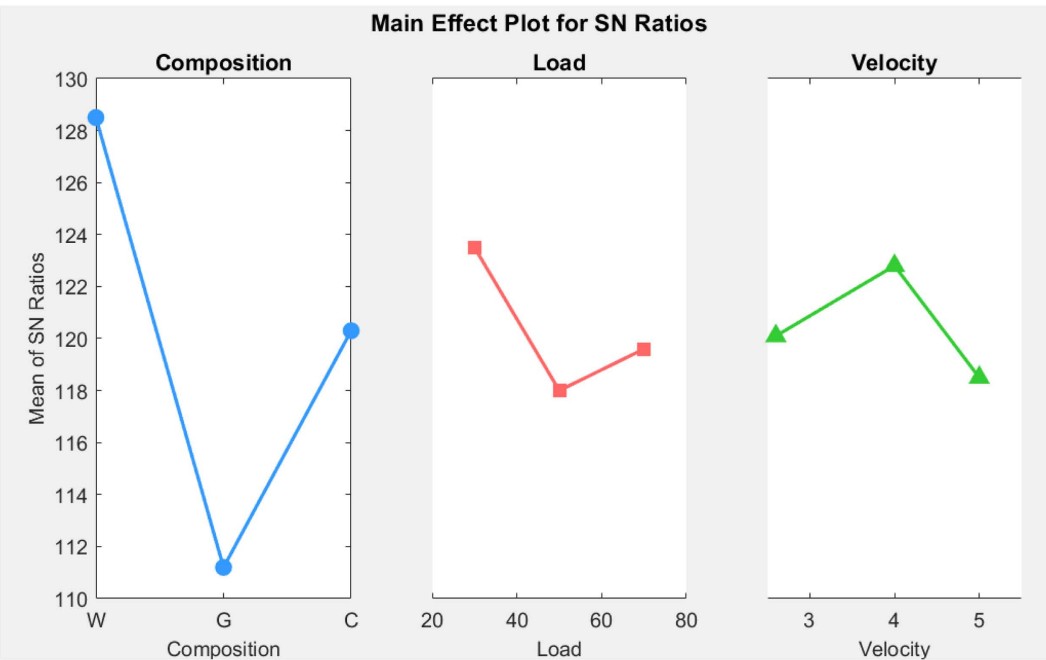

**Fig 5. Main effects plot for specific wear rate.**

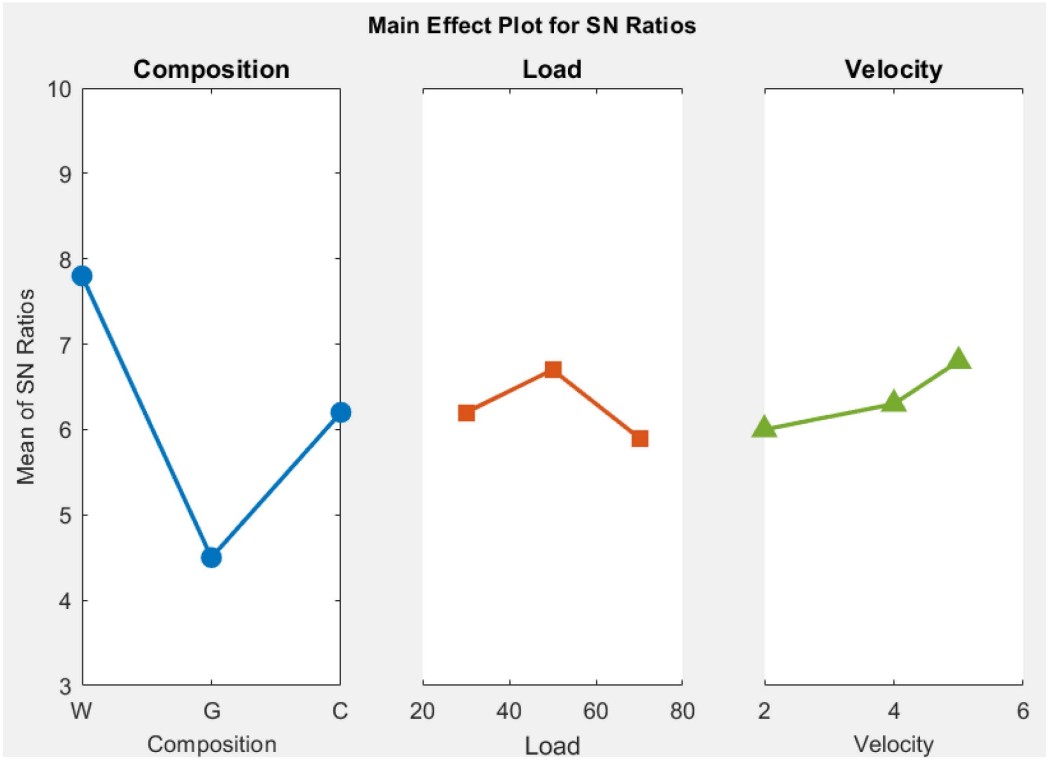

**Fig 6. Main effect plot for coefficient of friction.**

2 (111.1) and Level 3 (128.5), and the value of Level 1 (119.8) is between Level 2 and Level 3, which shows a positive progression. The level of load was highest for Level 1 (123.1) and was slightly less for Level 3 (119.8) and Level 2 (118.2) so that Level 1 had the most load of the three levels. Velocity was lower from Level 1 (119.8) to Level 3 (118.5), with the smallest distance between the three parameters for each of their respective levels was for velocity. The delta values indicate that the delta difference between the levels of composition values (17.4) was largest difference, followed by load (4.9) and then velocity (3.8). A line graph depicting data visualization of trends would be helpful to visualize the three parameters plotted individually and the values that represent the respective level of differences for each category. For comparative purposes, a graph would visualize the difference in parameters values amongst Level 1, Level 2, and Level 3, and provide visual representation of trends.

What does the Main Effect Plot for the value of the coefficient of friction (COF) show? It shows how makeup, load, and motion affect the coefficient of friction. From the table, it's clear that the "Composition" factor has the biggest change value (3.3) compared to the others, which means it is more variable. The load (0.8) or velocity (0.9) factors come in next. Based on each factor's delta value, the ranking of factors for influence on COF is composition, velocity then load. The S/N ratio of the COF also ranks equally, with composition having the greatest influence in both cases. The effect is greater in the case of composition and so it is plotted (i.e., compositional lines/bars) from a steeper angle. Thus the implication of these results is that composition is the most important factor to consider, when opportunity arises for optimization of friction performance.

The things that affect the Special Wear Rate (SWR) or Coefficient of Friction (COF) are shown in Tables 7 and 8, in order of how much they matter. In this instance, the levels of contribution identified for each factor are very similar to what has been identified in the ANOVA Tables 5 and 6. The fact that there is congruence within the relationship of the Taguchi design findings against the ANOVA findings goes to support the validity of the study. Additionally, the best contributing factors for the SWR and COF, have been compiled in Table 8. The best factors presented here are now considered key factors to underpin future studies, where the performance of the material or system of interest can be enhanced.

What Table 6 and Table 8 mean shows how makeup, load, and speed affect the signal-to-noise (S/N ratio) levels based on particular wear rate and coefficients of friction, respectively. Table 7 shows that makeup has the highest Delta value (17.4) for the specific wear rate. This means it is the most important factor, ranking first in terms of effect. This shows that the makeup you choose has a big impact on lowering the specific wear rate. The load comes next with a Delta of 4.9, followed by velocity with a Delta of 3.8, making them the second and third most critical factors, respectively. Table 7

**Table 7. Response for single-to-noise ratio for specific wear rate.**

| Level | Composition | Load | Velocity |
|---|---|---|---|
| 1 | 128.5 | 123.1 | 119.8 |
| 2 | 119.8 | 118.2 | 122.3 |
| 3 | 111.1 | 119.8 | 118.5 |
| Delta | 17.4 | 4.9 | 3.8 |
| Rank | 1 | 2 | 3 |

**Table 8. response for single-to-noise ratio for coefficient of friction.**

| Level | Composition | Load | Velocity |
|---|---|---|---|
| 1 | 7.9 | 6.2 | 5.9 |
| 2 | 4.6 | 6.7 | 6.2 |
| 3 | 6.3 | 5.9 | 6.8 |
| Delta | 3.3 | 0.8 | 0.9 |
| Rank | 1 | 3 | 2 |

**Table 9. Optimum factors as per main effect plots for S/N ratio.**

| Parameter | Composition | Load | Velocity |
|---|---|---|---|
| SWR | WPRC | 30N | 3.9 |
| COF | WPRC | 50N | 5.2 |

analyzes the coefficient of friction, where composition again shows the highest Delta (3.3), establishing it as the most dominant factor, ranked first. However, for the coefficient of friction, velocity is the second most influential factor with a Delta of 0.9, and load has the least effect with a Delta of 0.8. In both cases, composition plays a critical role, followed by load and velocity, though their rankings vary slightly between the two performance metrics. This emphasizes the need to carefully select composition to optimize both wear rate and friction.

In Table 9, the best factors for the Signal-to-Noise (S/N) ratio are shown based on main effect charts. The best conditions for certain performance measures have been found to be a load of 30N, a speed of 3.9 m/s, and a composition of WPRC. The best conditions for a coefficient of friction (COF) are a composition of WPRC, a load of 50N, and a speed of 5.2 m/s. These results show how important load and speed are for getting the best performance from the WPRC makeup.

## 4. Conclusion

1. The polymer friction composites reinforced with natural modifiers like walnut, coconut, and groundnut exhibit competitive tribological properties, with COF values ranging between 0.481 and 0.486, comparable to semi-metallic and ceramic pads.

2. Despite promising tribological performance, the composites showed limitations in thermal resistance and structural strength during high-load, high-temperature conditions, highlighting areas for improvement.

3. Accelerated wear tests indicated good wear resistance but the absence of long-term testing restricts the comprehensive assessment of their durability and reliability under real-world automotive conditions.

4. To enhance the development of these natural composites, future work must focus on optimization strategies for improved thermal stability and mechanical integrity, ensuring the materials perform well under fluctuating temperatures and repeated braking cycles.

5. Finally, to evaluate economic viability and large-scale production, it is crucial to conduct field trials, establish standardized testing protocols, and compare the composites' performance and cost-effectiveness against existing commercial materials.

6. Future studies should focus on evaluating the long-term durability of polymer composites under extreme conditions like humidity and thermal cycles, while developing scalable production methods and eco-friendly manufacturing techniques for high-performance applications.

## Author contributions

**Conceptualization:** Mitali Gore.

**Data curation:** Mitali Gore, Ajit Bhosale.

**Formal analysis:** Mitali Gore, Ajit Bhosale, Mithul Naidu, Robert Čep.

**Funding acquisition:** Robert Čep.

**Investigation:** Mithul Naidu.

**Methodology:** Mitali Gore, Ajit Bhosale, Mithul Naidu, Emad Abouel Nasr.

**Project administration:** Mitali Gore, Emad Abouel Nasr.

**Resources:** Sachin Salunkhe, Mitali Gore, Ajit Bhosale, Mithul Naidu.

**Software:** Ajit Bhosale, Mithul Naidu, Robert Čep, Emad Abouel Nasr.

**Supervision:** Sachin Salunkhe, Mithul Naidu, Robert Čep, Emad Abouel Nasr.

**Validation:** Sachin Salunkhe, Ajit Bhosale, Robert Čep, Emad Abouel Nasr.

**Visualization:** Sachin Salunkhe, Mitali Gore, Mithul Naidu, Robert Čep.

**Writing – original draft:** Sachin Salunkhe, Mitali Gore, Ajit Bhosale, Mithul Naidu, Robert Čep, Emad Abouel Nasr.

**Writing – review & editing:** Sachin Salunkhe, Ajit Bhosale, Emad Abouel Nasr.

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
