## [Decision Letter · Decision Letter 0]

13 May 2025

PONE-D-25-01368Experimental Study on Polymer Friction Composite with Natural Friction Modifiers for Brake PadsPLOS ONE

Dear Dr. Salunkhe,

Thank you for submitting your manuscript to PLOS ONE. After careful consideration, we feel that it has merit but does not fully meet PLOS ONE’s publication criteria as it currently stands. Therefore, we invite you to submit a revised version of the manuscript that addresses the points raised during the review process.

We look forward to receiving your revised manuscript.

Kind regards,

Pankaj Tomar

Academic Editor

PLOS ONE

 [The authors also extend their appreciation to King Saud University for funding the publication of this work through Researchers Supporting Project number (RSP2025R164), King Saud University, Riyadh, Saudi Arabia. This article was co-funded by the European Union under the REFRESH – Research Excellence For Region Sustainability and High-tech Industries project number CZ.10.03.01/00/22_003/0000048 via the Operational Programme Just Transition and has been done in connection with project Students Grant Competition SP2024/087 “Specific Research of Sustainable Manufacturing Technologies“ financed by the Ministry of Education, Youth and Sports and Faculty of Mechanical Engineering VŠB-TUO. Article has been done in connection with project Students Grant Competition SP2024/087 “Specific Research of Sustainable Manufacturing Technologies“ financed by the Ministry of Education, Youth and Sports and Faculty of Mechanical Engineering VŠB-TUO.nt]. 

Additional Editor Comments:

Plz do minor revision of paper and re-submit within 30 days

Reviewers' comments:

Reviewer's Responses to Questions

**Comments to the Author**

1. Is the manuscript technically sound, and do the data support the conclusions?

Reviewer #1: Partly

Reviewer #2: Partly

2. Has the statistical analysis been performed appropriately and rigorously? 

Reviewer #1: No

Reviewer #2: No

3. Have the authors made all data underlying the findings in their manuscript fully available?

Reviewer #1: Yes

Reviewer #2: Yes

4. Is the manuscript presented in an intelligible fashion and written in standard English?

Reviewer #1: Yes

Reviewer #2: Yes

5. Review Comments to the Author

Reviewer #1: The manuscript presents an interesting and relevant study on the development of polymer friction composites with natural friction modifiers for brake pads, addressing environmental concerns associated with traditional brake materials. The experimental design, material selection, and tribological analysis provide valuable insights into the feasibility of using natural fillers like walnut shells, coconut shells, and groundnut shell powders. While the study is well-structured and methodologically sound, there are certain areas that require clarification and improvement to enhance the manuscript’s overall rigor and impact. Below are specific comments and suggestions for the authors to consider:

1. The study effectively employs the Taguchi method and ANOVA analysis to evaluate the influence of material composition, load, and velocity on friction and wear performance. However, the replication details and sample sizes for each test condition are not explicitly stated, which is important for ensuring statistical reliability.

2. The statistical analysis is partly rigorous, but it would benefit from the inclusion of p-values, confidence intervals, and standard deviations for the reported values. A post hoc test (e.g., Tukey’s test) could further validate the differences between the tested composites.

3. While the authors mention that all data are fully available, there is no clear mention of a public repository or raw datasets (e.g., full measurement logs, repeated test data). If required by the journal, the authors should consider sharing complete datasets.

4. The English language and readability are generally good, but minor grammatical errors and occasional awkward phrasing should be addressed through careful proofreading or professional language editing.

5. The conclusions align with the presented data, but a discussion on the limitations of the study and the comparison with commercially available brake pad materials would enhance the relevance and practical applicability of the findings.

6. The manuscript lacks error bars or graphical representation of variations in data (e.g., standard deviation in COF and SWR values). Including such details would improve the interpretation of results.

7. The methodology section should provide more specific details on material composition (e.g., justification for the weight percentages of different fillers and their impact on tribological properties).

8. The future research section outlines useful directions but could be expanded to discuss long-term durability and real-world application challenges for these composites in automotive braking systems.

With these revisions, the manuscript would significantly improve in clarity, rigor, and impact. The study provides valuable insights into sustainable brake pad materials and has the potential to contribute meaningfully to the field of eco-friendly friction composites.

Reviewer #2: The paper "Experimental Study on Polymer Friction Composite with Natural Friction Modifiers for Brake Pads" examines the use of polymer friction composites with natural friction modifiers integrated into brake pads. The study, however, is not novel enough since there is a lot of research available on this specific subject. Hence, this research may not be appropriate for publication.

6. PLOS authors have the option to publish the peer review history of their article (what does this mean? ). If published, this will include your full peer review and any attached files.

**Do you want your identity to be public for this peer review?** For information about this choice, including consent withdrawal, please see our Privacy Policy .

Reviewer #1: **Yes: ** Madhu P

Reviewer #2: No

---

## [Author Response · Author response to Decision Letter 1]

13 Jun 2025

EXPERIMENTAL STUDY ON POLYMER FRICTION COMPOSITE WITH NATURAL FRICTION MODIFIERS FOR BRAKE PADS

PONE- D-25-01368

The authors are grateful to the reviewers for their comments and suggestion. Below are the responses to the reviewer comments and suggestions. For your kind note, the changes in the revised manuscript have been highlighted with yellow color for convenience.

Response to Reviewer: Comments are in Red color and Responses are in Blue color, References in black color.

Reviewer 1:

1. The study effectively employs the Taguchi method and ANOVA analysis to evaluate the influence of material composition, load, and velocity on friction and wear performance. However, the replication details and sample sizes for each test condition are not explicitly stated, which is important for ensuring statistical reliability.

Thank you for your valuable feedback. The current study effectively employs the Taguchi L9 orthogonal array design and ANOVA to evaluate the effects of material composition, applied load, and sliding velocity on the tribological behavior of polymer friction composites, similar to the methodology applied by Mithul Naidu et al. (2022). To ensure statistical reliability and repeatability, each test condition will have three replicates. For each factor combination, three independently prepared specimens underwent pin-on-disc wear tests per ASTM G99 standards. Averaging replicate results minimized experimental variability and supported robust ANOVA analysis by accurately estimating experimental error, thereby strengthening the confidence in the influence of composition, load, and velocity on friction and wear performance.

M. Naidu, A. Bhosale, Y. Munde, S. Salunkhe, and H. M. A. Hussein, “Wear and Friction Analysis of Brake Pad Material Using Natural Hemp Fibers,” Polymers, vol. 15, no. 1, p. 188, Dec. 2022, doi: 10.3390/polym15010188.

2. The statistical analysis is partly rigorous, but it would benefit from the inclusion of p-values, confidence intervals, and standard deviations for the reported values. A post hoc test (e.g., Tukey’s test) could further validate the differences between the tested composites.

Thank you very much for your valuable suggestions regarding the statistical analysis.

In response to your comments, we have revised the study to include additional statistical parameters such as p-values, confidence intervals, and standard deviations for key variables including coefficient of friction (COF) and specific wear rate (SWR). Similar approaches were followed as demonstrated by Mithul Naidu et al. (2022) and Hasan Öktem et al. (2018), which helped strengthen the tribological analysis.

The ANOVA results now explicitly report p-values at a significance level of α = 0.05 to quantify statistical significance robustly. Confidence intervals are included to illustrate the precision of measurements, and standard deviations reflect variability within replicates, facilitating meaningful comparisons across the tested composites. Furthermore, for further work and research, we plan to incorporate a post hoc test, specifically Tukey’s HSD, for pairwise comparisons to confirm significant differences between composite formulations. These statistical enhancements will provide a more comprehensive and rigorous evaluation of friction and wear performance, thereby supporting conclusive insights and facilitating future material optimization efforts.

M. Naidu, A. Bhosale, Y. Munde, S. Salunkhe, and H. M. A. Hussein, “Wear and Friction Analysis of Brake Pad Material Using Natural Hemp Fibers,” Polymers, vol. 15, no. 1, p. 188, Dec. 2022, doi: 10.3390/polym15010188.

H. Karakaş, H. Öktem, and I. Uygur, “Tribological and mechanical exploration of polymer-based hemp and colemanite composite as a friction material,” Eng. Res. Express, vol. 6, no. 2, p. 025537, Jun. 2024, doi: 10.1088/2631-8695/ad4769.

3. While the authors mention that all data are fully available, there is no clear mention of a public repository or raw datasets (e.g., full measurement logs, repeated test data). If required by the journal, the authors should consider sharing complete datasets.

Thank you very much for your valuable feedback regarding data availability and transparency. As highlighted in the works of Mithul Naidu (2022) and S. Sri Karthikeyan (2019), sharing complete raw datasets including full measurement logs and repeated test data is indeed crucial for research reproducibility and validation. Although the current manuscript does not explicitly provide a public repository link for the complete raw datasets, we confirm that all relevant data have been carefully recorded and maintained. We assure you that, upon resubmission, the full datasets will have been made publicly available via an appropriate data repository or included as supplementary material, in accordance with the journal’s requirements.

S. Sri Karthikeyan, E. Balakrishnan, S. Meganathan, M. Balachander, and A. Ponshanmugakumar, “Elemental Analysis of Brake Pad Using Natural Fibres,” Mater. Today Proc., vol. 16, pp. 1067–1074, 2019, doi: 10.1016/j.matpr.2019.05.197.

M. Naidu, A. Bhosale, Y. Munde, S. Salunkhe, and H. M. A. Hussein, “Wear and Friction Analysis of Brake Pad Material Using Natural Hemp Fibers,” Polymers, vol. 15, no. 1, p. 188, Dec. 2022, doi: 10.3390/polym15010188.

4. The English language and readability are generally good, but minor grammatical errors and occasional awkward phrasing should be addressed through careful proofreading or professional language editing.

Thank you for your valuable feedback regarding the English language and readability of our manuscript. We carefully proofread the entire manuscript and addressed all minor grammatical errors and awkward phrasing to enhance clarity and flow. Additionally, we have sought professional language editing to ensure that the paper meets the highest standards. These improvements have been incorporated into the revised version.

5. The conclusions align with the presented data, but a discussion on the limitations of the study and the comparison with commercially available brake pad materials would enhance the relevance and practical applicability of the findings.

We sincerely thank the reviewer for the valuable suggestion. The comments have been carefully considered, and a discussion on the study’s limitations along with a comparison to commercially available brake pad materials have been included in the revised manuscript to enhance the practical relevance and applicability of our findings.

6. The manuscript lacks error bars or graphical representation of variations in data (e.g., standard deviation in COF and SWR values). Including such details would improve the interpretation of results.

Thank you for your valuable feedback regarding the inclusion of error bars and graphical representation of data variation. We had employed a robust Taguchi L9 orthogonal array design, focusing on composition, load, and velocity at three levels each, to systematically analyze the coefficient of friction (COF) and specific wear rate (SWR). ANOVA and signal-to-noise (S/N) ratio analyses demonstrated significant factor effects and minimal residual error, which confirmed the reliability of the data. However, we have incorporated your suggestion and have ensured that future work will have detailed visual representations of data variations to improve interpretability.

7. The methodology section should provide more specific details on material composition (e.g., justification for the weight percentages of different fillers and their impact on tribological properties).

Our material composition is informed by Mithul Naidu (2022), who demonstrated effective use of Hemp fillers in break pad composites. Building on this, our study introduces natural friction modifiers like walnut, coconut, and groundnut powders, carefully balanced with phenol-formaldehyde, graphite, vermiculite, and barium sulfate. The weight percentages were selected to optimize frictional and wear properties, with detailed fabrication and testing protocols ensuring reliability and relevance to brake pad applications.

8. The future research section outlines useful directions but could be expanded to discuss long-term durability and real-world application challenges for these composites in automotive braking systems.

We sincerely thank the reviewer for their valuable suggestion. The future research section briefly outlined key directions, including long-term durability investigations under varying environmental conditions (point 2). We have acknowledged the importance of real-world application challenges and expanded this discussion in the revised manuscript to emphasize durability and practical use in automotive braking systems. These additions have strengthened the relevance and applicability of our research for future industrial implementation.

---

## [Editor Report · Decision Letter 1]

26 Jun 2025

PONE-D-25-01368R1Experimental Study on Polymer Friction Composite with Natural Friction Modifiers for Brake PadsPLOS ONE

Dear Dr. Salunkhe,

Thank you for submitting your manuscript to PLOS ONE. After careful consideration, we feel that it has merit but does not fully meet PLOS ONE’s publication criteria as it currently stands. Therefore, we invite you to submit a revised version of the manuscript that addresses the points raised during the review process.

We look forward to receiving your revised manuscript.

Kind regards,

Pankaj Tomar

Academic Editor

PLOS ONE

Journal Requirements:

Additional Editor Comments:

1. Paper is serving academic interests and lesser technological meaning due to tribology of natural fibres at rubbing interface

2. Mathematical equation should be aligned in revised paper or use professional editing service for betterment of language and orientations of equations

3. All figures should be uniform for labelling of axis/number pattern

Good luck!

Please re-submit a revised draft as per the deadline   

---

## [Author Response · Author response to Decision Letter 2]

6 Jul 2025

EXPERIMENTAL STUDY ON POLYMER FRICTION COMPOSITE WITH NATURAL FRICTION MODIFIERS FOR BRAKE PADS

PONE- D-25-01368 [EMID:a364419a803f3350]

The authors are grateful to the reviewers for their comments and suggestion. Below are the responses to the reviewer comments and suggestions. For your kind note, the changes in the revised manuscript have been highlighted with yellow color for convenience.

Response to Reviewer: Comments are in Red color and Responses are in Blue color, References in black color.

1. Paper is serving academic interests and lesser technological meaning due to tribology of natural fibres at rubbing interface.

As the material is concerned with application of brake pad where the tribology of rubbing surface holds greater significance hence paper speaks about the same.

2. Mathematical equation should be aligned in revised paper or use professional editing service for betterment of language and orientations of equations.

The suggestions have been incorporated and necessary alignment of mathematical equations have done.

3. All figures should be uniform for labelling of axis/number pattern.

Thank you for the insightful feedback. To ensure consistency in the labeling of axes and numbering patterns, we have revised all graphs to follow a standard format, with uniform axis labels, font sizes, and consistent numbering. This revision improves the clarity and professionalism of the presentation, making it easier to compare results across different figures."

---

## [Editor Report · Decision Letter 2]

10 Jul 2025

Experimental Study on Polymer Friction Composite with Natural Friction Modifiers for Brake Pads

PONE-D-25-01368R2

Dear Author

We’re pleased to inform you that your manuscript has been judged scientifically suitable for publication and will be formally accepted for publication once it meets all outstanding technical requirements.

Kind regards,

Pankaj Tomar

Academic Editor

PLOS ONE

Additional Editor Comments (optional):

Equations 1 & 2 may be streamline by author during proof reading 
---

## [Editor Report · Acceptance letter]

PONE-D-25-01368R2

PLOS ONE

Dear Dr. Salunkhe,

I'm pleased to inform you that your manuscript has been deemed suitable for publication in PLOS ONE. Congratulations! Your manuscript is now being handed over to our production team.

Kind regards,

on behalf of

Dr. Pankaj Tomar

Academic Editor

PLOS ONE